# The Association between Blood Β-Hydroxybutyric Acid Concentration in the Second Week of Lactation and Reproduction Performance of Lithuanian Black and White Cows

**DOI:** 10.3390/ani12040481

**Published:** 2022-02-15

**Authors:** Indrė Mečionytė, Giedrius Palubinskas, Lina Anskienė, Ramūnas Antanaitis, Ayhan Yilmaz, Ilma Tapio, Vytuolis Žilaitis

**Affiliations:** 1Department of Animal Breeding, Veterinary Academy, Lithuanian University of Health Sciences, Tilžės Str. 18, 47181 Kaunas, Lithuania; giedrius.palubinskas@lsmuni.lt (G.P.); lina.anskiene@lsmuni.lt (L.A.); 2Large Animals Clinic, Veterinary Academy, Lithuanian University of Health Sciences, Tilžės Str. 18, 47181 Kaunas, Lithuania; ramunas.antanaitis@lsmuni.lt (R.A.); vytuolis.zilaitis@lsmuni.lt (V.Ž.); 3Department of Animal Science, Agriculture Faculty, Siirt University, 56100 Siirt, Turkey; ayilmaz@siirt.edu.tr; 4Genomics and Breeding, Production Systems, Natural Resources Institute Finland (Luke), Myllytie 1, 31600 Jokioinen, Finland; ilma.tapio@luke.fi

**Keywords:** BHB, lactation, season, milk yield, fertility

## Abstract

**Simple Summary:**

Determination of BHB concentration in the second week of lactation (WK 2) may allow us to predict the fertility properties of cows and help better manage farms. BHB concentration can be considered as a predictor trait of reproduction success. High BHB concentration requires a higher amount of insemination. The season in which the cows calve and the parity must be considered in the assessment as these factors affect BHB concentration in WK 2.

**Abstract:**

Hyperketonemia is a very common metabolic state in dairy cows, which result in lower milk production, impaired fertility, and increased frequency of other diseases. In this study, we aimed to determine the influence of season, parity, and milk yield of cows on beta-hydroxybutyrate (BHB) concentration in the second week of lactation (WK 2) and establish the relationship between BHB concentration in WK 2 and reproduction performance traits such as insemination rate and first insemination day of Lithuanian Black and White dairy cows. The study included clinically healthy Lithuanian Black and White cows (*n* = 692). Blood BHB concentration was measured using capillary blood samples collected after morning milking when cows were 7–10 DIM. The impact of WK 2 blood BHB concentration on the insemination rate and first insemination day were investigated. The effect of BHB was evaluated according to the season, parity, and milk yield per lactation (305 DIM). Significant differences were observed in BHB concentration in WK 2 due to season and parity, but no statistically significant differences were observed for milk yields (305 d). Increased blood BHB concentration in WK 2 negatively affected insemination rate (*p* < 0.001) and first insemination day (*p* < 0.001). The study findings indicate that BHB concentration in WK 2 depends on season and parity, while the milk yield is not associated with BHB concentration. High BHB concentration in WK 2 increases insemination rate and delays the first insemination day for high milk-yielding Lithuanian Black and White dairy cows.

## 1. Introduction

Metabolic load in high milk-yielding cows is a vital issue, and periodic changes are susceptible to a variety of metabolic and infectious diseases during transition processes (the period between three weeks before and three weeks after parturition) [1,2,3]. Massive mobilization of nonesterified fatty acids before, during, and after calving is a common metabolic feature in high-producing dairy cattle and is also an adaptive process for this new metabolic status. Β-hydroxybutyrate (BHB) is the ketone body that is increased in cow’s blood during early lactation due to negative energy balance (NEB). After calving, there is a tendency to use body reserves due to increased metabolic demands in high milk-yielding cows. Researchers have reported detrimental effects of NEB on metabolic processes, milk yield per lactation, and the immune system, along with other health problems [4,5,6]. Hyperketonemia (HYK) impairs the health of dairy cows by increasing the risk of onset of other early lactation diseases and negatively affects the reproductive status after calving. Walsh et al. [7] observed significant differences between cows with and without HYK in terms of reproduction performance. Likewise, Rutherford et al. [8] reported that high milk-yielding cows need a positive energy balance for reproductive success. Moreover, there may be an association between reproduction performance and milk yield of cows with HYK. In cows with low milk yield levels, HYK was found to be associated with lower risk of pregnancy to first insemination [9]. Moderate blood BHB levels during the early postpartum period affect response to the hormonal estrus synchronization protocol [10]. Postpartum return to estrus is associated with high concentration of estradiol and low concentration of BHB in early lactation [11]. In addition, other factors, such as the season of calving, breed, and management, should be considered for diagnosis of HYK in dairy cattle [12]. Daily energy balance may be the best indicator of metabolic load in dairy cows, but this is limited in animal breeding practice [13]. It requires the use of different and easy indirect indicators, such as concentration of nonesterified fatty acids (NEFA) and BHB in the blood. For diagnosis of hyperketonemia in dairy cattle, blood BHB concentration is one of the markers as it is a predominant circulating ketone body in ruminants [14]. It has been found that measurement of BHB is a useful diagnostic tool for prediction of the health of cows [15].

Lithuanian Black and White cows correspond to the model of global high-yielding cow breed. Comprising 70% of all cattle in Lithuania, they are an old native dairy cattle breed with significant regional and cultural value. Therefore, this study and its results are relevant for not just researchers but also for farmers and breeders of cows with high productivity [16].

The aim of this study was to explore the role of season, parity, and last lactation milk yield of cows on BHB concentration in WK 2 and to evaluate the relationship between BHB concentration and reproductive performance measures such as insemination rate and first insemination day in high-yielding Lithuanian Black and White cows.

## 2. Materials and Methods

This study was carried out in Lithuania on a dairy cattle farm of 1200 dairy cows kept in a loose housing system and located at (54.436224, 23.246784) according to the World Geodetic System from October 2018 to September 2019. The experiment was carried out in a geographic place determined as humid continental climate by W. Köppen [17], also known as hemiboreal climate. The average temperature isotherm is −4.5 °C in January and 17.5 °C in July. The average peak temperature for the months of June, July, and August is 20.1 °C. The average temperature for the months of December, January, and February is −3.6 °C. The detected humidity is lowest in winter months (December, 66–77%) and highest in summer months (May, 81–91%). 

Seasonal effects were examined by examining the concentration of BHB in WK 2 in spring, summer, autumn, and winter.

In total, 692 Lithuanian Black and White fresh dairy cows were selected for this experiment. The cows had 1–8 lactations and were clinically healthy (without any clinical sign of metritis, lameness, mastitis, displaced abomasum, or indigestion, with an average rectal temperature of 38.8 °C). Cows were milked three times per day. The average milk production during last lactation (305 DIM) was 9613 ± 39.97 kg per cow. All the milked cows were fed ad libitum with the same total mix ratio twice a day (06:00 and 18:00 h). Rations were calculated on the “HYBRIMIN Futter 2008” system, which is based on the German approved DLG standards [18]. The diet was formulated to meet the requirements of a 650 kg Lithuanian Black and White cow producing 50 kg/d (4.2% fat, 3.5% proteins). The chemical composition of the ration was as follows: 27.1 kg DM (of total ration), 174 g crude protein (per kg of DM), 258 g nonfiber carbohydrates (per kg of DM), 42 g crude fat (per kg of DM), and 160 g crude fiber (per kg of DM). The total energy for lactation (NET) was 7.01 MJ/kg DM and metabolizable energy (ME) was 11.52 MJ/kg DM. The total mixed ration consisted of 4.7 kg DM corn silage (17.2%), 8.13 kg DM grass silage (29.8%), 3.2 kg DM crushed corn grains (11.7%), 1.37 kg DM sugar beet pulp silage (5.0%), 8.5 kg DM grain concentrate mash (31.1%), 0.36 kg DM wheat straw (1.31%), and 0.77 kg DM molasses (2.8%). The body condition score of the calving cows was in the range of 3.5–3.75 (on a 5-point scale) [19,20].

The blood samples for BHB concentration determination were collected on strips from the ear at 7–10 DIM in the mornings after milking before feeding. The Nova Vet Blood Ketone/Glucose Meter device (Taiwan) and Nova Vet Glucose Test Strips were used to determine blood BHB concentration in WK 2.

Cows were divided into groups according to the season in which they calved:Spring—196 cows,Summer—79 cows,Autumn—188 cows, andWinter—229 cows.

Cows were divided into groups according to parity on which the experiment was started: first—195 cows, second—178, third—148, fourth—86 cows, fifth—38 cows, sixth—25, seventh—14, and eighth—8.

Reproductive outcomes were collected using the GEA herd management program (Germany). Cows were divided into groups according to insemination rate: 1—333 cows, 2—171 cows, 3—118 cows, 4—50 cows, 5—11 cows, and 6—9 cows.

The statistical analysis of data was performed using the SPSS 25.0 software package (IBM Corp. 2017. IBM SPSS Statistics for Windows, Armonk, NY, USA). Normal distributions were assessed using the Kolmogorov–Smirnov test. The results are presented as the mean ± standard error of the mean (M ± SE). Duncan’s multiple range test (DMRT) was used to analyze the differences in mean values of normally distributed variables. A probability of less than 0.05 was deemed reliable (*p* < 0.05). The Pearson correlation (r) was determined to define the linear relationship between BHB concentration in WK 2 and season, parity, first insemination time, insemination rate, and milk production. In this study, the regression method was used to measure the effects of independent variables on the dependent variable: insemination rate, first insemination day, and blood BHB concentration in WK 2.

## 3. Results

### 3.1. Descriptive Results of the Herd

The average BHB concentration in WK 2 of all cows was 0.79 ± 0.30 nmol/L. The lowest (0.59 nmol/L) BHB concentration was on the 65th DIM and highest (1.14 nmol/L) on the 105th DIM. The average milk yield during last lactation (305 d) of all cows was in the range of 7008—13,669 kg. The average first insemination day was 81.40 ± 3.77 days, and the insemination rate of all cows was 1.98 ± 1.06 times.

### 3.2. Effect of Season and Parity on BHB Concentration in WK 2

The effect of the season on blood BHB concentration in WK 2 was statically significant (Figure 1A). The mean BHB concentration in WK 2 was 15.73% higher in summer compared to spring (*p* < 0.001) and 13.48% higher in summer compared to winter (*p* < 0.01), representing statistically significant differences. The mean BHB concentration in WK 2 was 6.74% higher in summer compared to autumn, which was a noticeable difference but not statistically significant (*p* > 0.05). The mean BHB concentration in WK 2 was 9.64% higher in autumn compared to spring (*p* < 0.05) and 7.23% higher in autumn compared to winter (*p* > 0.05). The difference (2.67%) between the mean BHB concentration in WK 2 in spring and winter was not statically significant.

We observed significant differences between mean BHB concentration in WK 2 and the parity of cows (Figure 1B). Significant differences were obtained between primiparous and multiparous cows for BHB concentration in WK 2, indicating that multiparous dairy cows had higher BHB concentration compared to primiparous ones (14.44%; *p* < 0.05). Statistically significant differences were estimated between the mean BHB concentration in different lactations (Table 1).

The other differences between the mean BHB concentration in WK 2 in different lactations were not statistically significant (Figure 1B).

### 3.3. The Relationship between Blood BHB Concentration in WK 2 and Milk Yield Per Last Lactation (305 d)

We observed significant differences between milk yield per lactation (305 d) based on parity (*p* < 0.001). We estimated the lowest milk yield on the eighth lactation, which was 23.06% lower compared to the highest in the fifth lactation (*p* < 0.001). High statistically significant differences (*p* < 0.001) were observed for the lowest milk yield of cows in the second lactation compared to the third, fourth, fifth, sixth, and seventh lactations (12.95%, 14.84%, 19.55%, 14.63%, and 16.64%, respectively). Despite the milk yield varying by parity, we did not find a statistically significant relationship between milk yield and BHB concentration in WK 2 (*p* > 0.05), as shown in Table 2.

### 3.4. The Effect of BHB Concentration on Reproductive Outcomes

We estimated negatively statistically significant influence of BHB concentration in WK 2 on insemination rate (Figure 2). Cows with insemination rate 1 had 22.22% lower BHB concentration in WK 2 than cows with insemination rate 4 (*p* < 0.001) and 9.41% lower than cows with insemination rate 3 (*p* < 0.05). Cows with insemination rate 2 had 24.24% lower BHB concentration in WK 2 compared to cows with insemination rate 4 (*p* < 0.001) and 11.76% lower than cows with insemination rate 3 (*p* < 0.05). Cows with insemination rate 3 had 14.14% lower BHB concentration in WK 2 than cows with insemination rate 4 (*p* < 0.01). We observed that as the blood BHB concentration in WK 2 increased, the insemination rate also had a tendency to increase (*y* = 0.0297x + 0.746, *R^2^* = 0.4109).

Higher BHB concentration in WK 2 was related to delayed first insemination day. The correlation coefficient between blood BHB concentration in WK 2 and first insemination day was weak positive but statistically significant (*r* = 0.194, *p* < 0.001), Figure 3.

We estimated a statistically significant relationship between season and insemination rate (*p* < 0.01), as shown in Table 3. Cows inseminated in autumn had the lowest mean insemination rate (14.85% lower compared to cows that were inseminated in spring and 20.37% lower than those inseminated in summer, *p* < 0.01). A statistically significant relationship was found between insemination rate and BHB concentration in WK 2 in spring (*p* < 0.01) and in summer and autumn (*p* < 0.05). Statistically significant mean differences were also detected between first insemination day in different seasons. Analysis showed that the first insemination was 6.93% earlier in spring compared to autumn and 6.35% earlier compared to winter (*p* < 0.001). A statistically significant relationship was also found between first insemination day and BHB concentration in spring (*p* < 0.001) and winter (*p* < 0.01).

The insemination rate was found to differ between different lactations (Table 4). The highest mean was detected in the sixth lactation (21.61% higher compared to the first lactation and 22.03% compared to the third lactation, *p* < 0.05). A statistically significant relationship was found between insemination rate and BHB concentration in WK 2 in the first lactation (*p* < 0.01) and in the third and fourth lactations (*p* < 0.05). No statistically significant mean differences were estimated between first insemination day in different lactations (*p* > 0.05). A statistically significant relationship was found between first insemination day and BHB concentration in WK 2 in the fourth (*p* < 0.001), third (*p* < 0.01), and first (*p* < 0.05) lactations.

## 4. Discussion

In our study, higher statistically significant difference in mean blood BHB concentration on second week of lactation (WK 2) of cows was detected in summer compared to spring and in summer compared to winter. A study in Iran with the Holstein breed of cows showed that the blood BHB concentration in cows can be affected by factors such as season of calving, breed, and herd management [21]. In the present investigation, the results revealed the effect of parity and season on prevalence of HYK. Previous studies with the Holstein breed of cows have detected greater prevalence of HYK in spring, but Vanholder et al. [22] and Santschi et al. [23] reported contrasting results on the effect of season on prevalence of HYK (late autumn and winter or summer). Antanaitis et al. [24] carried out research on Lithuanian Black and White cows and reported that milk BHB concentration had a tendency to increase with increasing number of lactations. Consequently, the highest level of BHB was estimated in the oldest cows, which is similar to our results. However, the authors also reported that BHB concentration in multiparous cows was higher than in primiparous cows, which is opposite to the findings of this study.

A study with Brown Swiss cows reported that multiparous cows use dietary energy more effectively compared to primiparous cows [25]. In the research by Piñeyrúa et al. [26] with Holstein cows, the high increase in BHB concentration of first lactation cows resulted from adaptation to a new technological process, and cows that could not adapt to the new status were removed from the herd. Multiparous cows have more adequate and suitable metabolism for the lactation process and have lower blood BHB concentration compared to primiparous cows. According to Roberts et al. [27] and McArt et al. [28], cows with high blood BHB concentration should be removed and culled from the herd in early lactation. Mohammed et al. [29] reported differences in relation to the number of lactations. Their results clearly revealed that prevalence of HYK in cows depends on the age of the cow or the calving month.

In the present study, we did not find a relationship between BHB concentration in WK 2 and milk yield (305 d). Moreover, no statistically significant differences were observed between milk yield (305 d) of primiparous and multiparous dairy cows and blood BHB concentration. Several studies have reported contradictory results, with researchers finding a relationship between production and blood or milk BHB concentration [19,28,30]. However, no significant differences were observed between cows with and without HYK by van der Drift et al. [31], Djoković et al. [32], and Chandler et al. [15]. The negative impact of HYK in cows of the Holstein breed was more pronounced in the first week compared to the second week of lactation [5,21]. Kayano and Kataoka [33] and Santschi et al. [23] reported that differences in milk yield between cows with and without HYK increased during lactation due to the cumulative NEB in hyperketonemic cows.

In this study, an increase in BHB concentration in WK 2 negatively affected the fertility traits of cows, including the insemination rate and first insemination day (DIM). We found a relationship between reproduction outcome and BHB concentration in WK 2 depending on the season (*p* < 0.01). The highest statistically significant correlation was detected between BHB concentration and reproduction outcomes in spring. We did not find an obvious relationship between BHB concentration in WK 2 and reproduction outcomes, but there was a statistically significant, weak correlation in the fourth, third, and first lactations. Therefore, BHB concentration in WK 2 should be analyzed in different seasons and lactations in order to increase reproduction outcomes. In a study with Ayrshire cows, Grohn et al. [34] also reported that prevalence of HYK increased with age, with the peak being observed in the third and fifth lactations.

Rutherford et al. [8] reported that animals with HYK had greater insemination rate, lower peak activity, shorter activity at estrus, and longer interval from calving to first observed estrus compared to animals without HYK. Walsh et al. [7] carried out research with Holstein cows and observed the significant differences in reproduction success between cows with and without HYK. The authors found that increased BHB levels in the first two weeks after calving negatively affected pregnancy status at first artificial insemination. These results can be considered to be in line with our results. On the other hand, studies by Chapinal et al. [35] and McArt et al. [28] with Holstein breed of cows did not show significant differences between animals with and without HYK in terms of reproduction successes.

The discussion shows that similar studies have been performed with other breeds of dairy cattle, but research findings in the context of Lithuanian Black and White breed of dairy cows is lacking.

## 5. Conclusions

According to the results of our study, we can conclude that BHB concentration of Lithuanian Black and White cows in WK 2 depends on the season and parity of cows. However, BHB concentration in WK 2 is not related to milk yield (305 d) in Lithuanian Black and White cows. Increased BHB concentration clearly impacts the fertility rates, with insemination rate and first insemination day having a statistically significant relationship with BHB concentration in WK 2 in different seasons and lactations of cows. Analysis of BHB concentration in WK 2 can be a predictor of reproductive performance traits.

## Figures and Tables

**Figure 1 animals-12-00481-f001:**
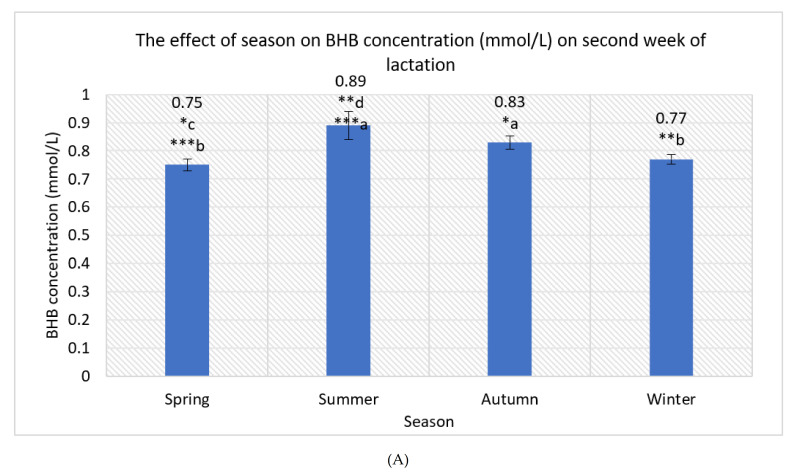
(**A**) The effect of season on mean values of BHB concentration on second week of lactation (mmol/L). a, b, c, d—marking with different letters indicates the mean differences between seasons (spring, summer, autumn, and winter). *** *p* < 0.001; ** *p* < 0.01; * *p* < 0.05. (**B**) The effect of parity on BHB concentration (mmol/L) on second week of lactation. a, b, c, d, e, f, g, h—marking with different letters indicates mean differences between the investigated indicators. *** *p* < 0.001; ** *p* < 0.01; * *p* < 0.05.

**Figure 2 animals-12-00481-f002:**
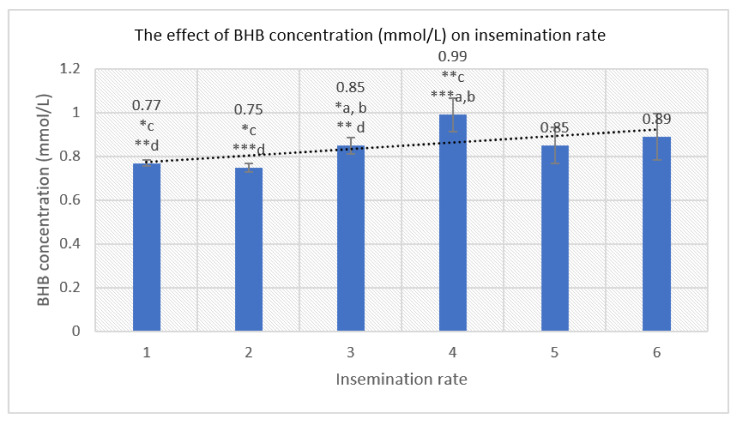
The relationship between mean BHB concentration (mmol/L) on second week of lactation and insemination rate. a–d—marking with different letters indicates the mean differences between the investigated indicators. *** *p* < 0.001; ** *p* < 0.01; * *p* < 0.05.

**Figure 3 animals-12-00481-f003:**
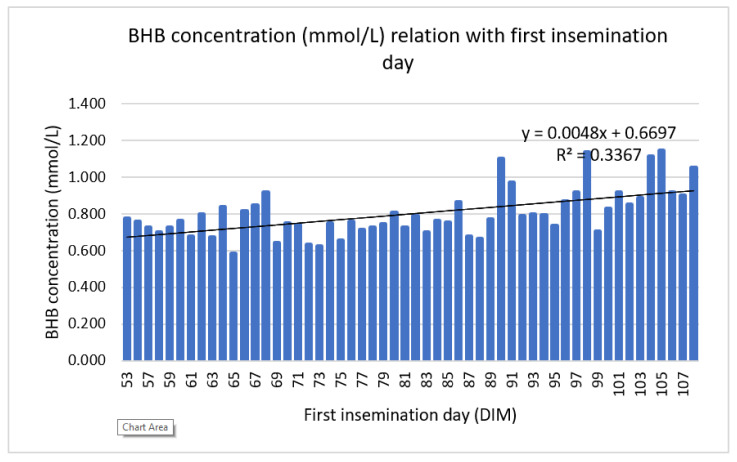
The effect of mean BHB concentration (mmol/L) on second week of lactation on first insemination day.

**Table 1 animals-12-00481-t001:** Statistically significant differences between lactations with higher and lower BHB concentration.

Lactations with Higher vs. Lower BHB Concentration	Different in Percents (%)	*p* Value (*p*<)
1 vs. 5	13.48	0.05
1 vs. 7	16.3	0.001
1 vs. 8	18.95	0.05
2 vs. 7	14.81	0.001
2 vs. 5	17.59	0.01
2 vs. 8	12.04	0.01
3 vs. 7	9.78	0.01
4 vs. 7	9.78	0.05
6 vs. 7	13.68	0.05

**Table 2 animals-12-00481-t002:** The results of milk yield by parity and relationship of the last lactation milk yield with BHB concentration in WK 2.

Parity	Milk Yield, kg (305 d)	Correlation of Milk Yield of Last Lactation (305 d) with BHB in WK 2 on Current Lactation	*p* Value of Correlation Coefficient
1	-	-	-
2	8717.10 ± 96.90 *** c,d,e,f,g	−0.004	0.963
3	10,013.87 ± 116.40 *** b,** e,h	0.055	0.505
4	10,235.76 ± 201.01 *** b, ** h, * e	0.039	0.725
5	10,834.76 ± 231.24 *** b,h, ** c, * d	−0.088	0.601
6	10,210.45 ± 307.53 *** b, ** h	−0.253	0.223
7	10,457.25 ± 430.99 *** b,* h	−0.195	0.505
8	8335.80 ± 1293.70 *** e, ** c,d	0.320	0.536

a–h—marking with different letters indicates mean differences between the investigated indicators. *** *p* < 0.001; ** *p* < 0.01; * *p* < 0.05.

**Table 3 animals-12-00481-t003:** The results of insemination rate and first insemination day in different seasons and relationship between insemination rate and BHB concentration and first insemination day and BHB concentration.

Season	Insemination Rate	Correlation of Insemination Rate with BHB in WK 2	First Insemination Day (DIM)	Correlation of First Insemination Day with BHB in WK 2	*p* Value of Correlation Coefficient
Spring	2.02 ± 0.09 **c	0.219 **	77.55 ± 0.89 ** b, *** c,d	0.296	0.001
Summer	2.16 ± 0.13 **c	0.241 *	82.49 ± 1.50 ** a	0.148	0.192
Autumn	1.72 ± 0.08 ** a,b	0.163 *	83.32 ± 1.06 *** a	0.135	0.065
Winter	1.92−0.07	0.076	82.81 ± 0.98 *** a	0.170	0.01

a–d—marking with different letters indicates the mean differences between seasons (spring, summer, autumn, and winter). *** *p* < 0.001; ** *p* < 0.01; * *p* < 0.05.

**Table 4 animals-12-00481-t004:** The results of insemination rate, first insemination day in different lactations, and relationship between insemination rate and BHB concentration and insemination day and BHB concentration.

Parity	Insemination Rate	Correlation of Insemination Rate with BHB in WK 2	*p* Value of Correlation Coefficient	First Insemination Day	Correlation of First Insemination Day with BHB in WK 2	*p* Value of Correlation Coefficient
1	1.85 ± 0.07 * f	0.219	0.01	81.18 ± 1.03	0.154	0.032
2	2.03 ± 0.09	0.033	0.663	81.28 ± 1.03	0.127	0.097
3	1.84 ± 0.09 * f	0.199	0.05	80.20 ± 1.18	0.227	0.006
4	1.91 ± 0.13	0.219	0.05	82.24 ± 1.55	0.360	0.001
5	1.92 ± 0.16	0.172	0.303	84.00 ± 2.57	0.216	0.193
6	2.36 ± 0.26 * a,c	−0.011	0.959	82.72 ± 2.89	−0.054	0.798
7	1.64 ± 0.23	0.264	0.362	82.93 ± 3.72	0.139	0.636
8	1.83 ± 0.31	0.255	0.625	86.17 ± 6.26	0.127	0.811

a–h—marking with different letters indicates mean differences between the investigated indicators. * *p* < 0.05.

## Data Availability

The research was conducted on a private farm (located by World Geodetic System (54.436224; 23.246784)) based on private data from that farm.

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
