# Peer review of "The Association between Blood Β-Hydroxybutyric Acid Concentration in the Second Week of Lactation and Reproduction Performance of Lithuanian Black and White Cows"

_animals, 2022, doi:10.3390/ani12040481_

Round 1
Reviewer 1 Report
Title
Maybe it is better to change the title to “The Association between Blood Β-Hydroxybutyric Acid Concentration, a Biomarker for Early Diagnosis of Hyperketonemia, and Postpartum Cow’s Production and Reproduction Performances”
Abstract
L14 I suggest to replace “frequent” with “common”
L16 Please change “vital challenge” to “a vital challenge”
L17 change “sub-clinical” to “subclinical”
L25 Generally the peak milk yield occurs at 45 to 100 days postpartum. The time of peak milk yield occurs later in primiparous cows than multiparous cows. In your study, you have both primiparous and multiparous cows, do you think d 45 represent the milk yield for both primiparous and multiparous cows?
L20 delete per cow
L29 delete “=”
L37 Delete “at”. As I mentioned earlier, the time of peak milk yield may differ between primiparous and multiparous cows. Therefore, you had better reanalyze the association of the concentration of BHB between milk yield/ peak milk yield in primiparous and multiparous cows individually.
L26-44 If the results are not significant, it is necessary to mention them.
Introduction
L70 change “which it is predominant” to “which is a predominant”
Materials and Methods
L81 Insert . between +38.8 ◦C) and cows.
L82-83 Diet chemical compositions should be listed in a table. This can be provided as a supplemental material. How many times were the cows milked daily?
L83 Total mixed ration only occurs in the whole manuscript once. It is not necessary to use its abbreviation. Please double check the whole manuscript.
L85-86 3.5-3.75 is the range of the body condition score but not the mean value. Please provide the mean using the format of means ± SE. Also please indicate which body condition evaluation system did you use (Literature should be provided). Did you check if body condition affects the concentration of BHB? High body condition cows are more likely to mobilize more fat resulting in higher BHB.
L87-88 Were blood samples collected before or after feeding?
L88 Add was between “BHB (mmol/L)” and “determined”.
Results
L116-119 I feel confused about the description of your results. Please rewrite the sentences and make it clear.
L119-121 I don’t see the significant differences in BHB caused by lactation number. Could you please put some symbols in Figure1 b to indicate the differences like what you did in Figure1 a? What does the Y and X axis represent individually?
Discussion
It is not necessary to show P values in the discussion section.
Author Response
hello,
thank you very much for your comments and remarks. We edited article according to your recommendations.

Reviewer 2 Report
Authors conducted a study of relationship between BHB levels and some reproductive and productive outcomes. The novelty of the study is low, because there are some recent and old studies regarding this topic, but the breed studied could be interesting. As authors have enough data (N = 692 cows) from this breed, I encourage authors to improve the manuscript in order to be suitable to publish in this journal. The main concerns are about language, style and processing data – results.
General comments:
- Authors should follow journal guidelines.
- A deep English revision is required.
- If authors just performed in 692 cows a BHB strip and collected the data, I feel insufficient the results to be published. They must justify why this research is important to be shared. In my opinion, it could be important if they emphasize on the breed and the needs of researching and sharing knowledge of this breed.
- In general, a complete improve and data work should be performed to understand the manuscript.
- Discussion and conclusions sections were not evaluated because I did not understand completely M&M, nor results sections. I apologize for the inconvenience.
Author’s information: please provide all author acronyms.
Simple summary: this section is missing.
Abstract: need to be shorten. It exceeds 200 words.
Ln 20. Provide mean ± standard deviation.
Ln 26 – 44. Please take care of journal format.
Ln 26. Rewrite. Like this: “… BHB concentration due to season and lactation number.”
Ln 29. Delete “=”
Ln 28–30. You just said it was different due season. Is spring different? Please explain.
Ln 28. P value in italic. Please, apply throughout the manuscript.
Ln 33. “fertility properties” changed it to “reproductive outcomes” or something similar.
Ln 36. The BHB data is for multiparous? Is the difference between heifers and multiparous? Specify.
In general, abstract need to be shorten, be formatted as journal guidelines, enhance clarity of results presented and include a conclusion.
Introduction:
- A deep language analysis must be performed.
- I would emphasize on differences with the Lithuanian Black and White cattle, because there is a lot of bibliography about BHB and the indexes you measured. So, focus the article on the impact on its breed and the importance of this study for these breeders.
- Expand, a little more, about outcomes and repercussion of BHB
Material and methods:
Ln 79. Please express appropriately the age and lact. Number.
Ln 81. Delete “+”, and use °C for Celsius degrees.
Ln 82. Express as said above.
Ln 84. Include NRC reference.
Ln 86. Include BCS reference.
Ln 87. Explain how you did perform the ear sampling.
Ln 92–96. Check journal guidelines. Maybe they should be mentioned as references.
Ln 97–109. Did you take into account the factor “animal” by itself? Please explain.
Questions:
- I found incomplete the M&M description.
- When did you perform the experiment? You described influence of seasonality in the abstract, how did you measure that? THI data?
- Describe how results are given in that BHB measurer
- How did you collect and process the reproductive outcomes?
- Please explain more about the herd and farm management.
Results:
Prior to the section 3.1 I would recommend authors to describe the descriptive results of the herd throughout the study.
Section 3.1:
- Please do not repeat results, if they appear in a graphic, do not describe them completely again.
- Graphics should be improved.
- I encourage authors to provide tables in stead of graphics. There we could see the N of each variable.
- Numbers always: 0.08, do no use comma
Section 3.2:
- The same comments than for 3.1
- Very difficult to follow.
- Insemination number in global, but depending on parity would be also welcome and interesting to share. As well, the interaction with season.
- Figure 2b. I do not understand
- Figure 3. What do you want to say with this graphic? Sorry, but I do not understand either.
Section 3.3:
- The same comments than for 3.1 and 3.2
- Very difficult to follow and controversial results are expressed.
- I recommend authors not to use the graphics used for Figure 4. They are difficult to read in this kind of data.
Discussion and conclusions:
- As the M&M and results sections are not completely understandable, I am not able to evaluate this both sections. I apologize for the inconvenience.
Author Response
Hello,
thank you very much for your comments and remarks. We edited article according to your recommendations.

Round 2
Reviewer 1 Report
Thanks for answering my questions! But I still can't understand the way you present your data including the diet you provided. Besides, English language editing is needed. Sorry I can't provide more comments unless you make your figures more clear and understandable to readers.
Author Response

(The authors gave the same response as above.)

Reviewer 2 Report
Authors did apply some comments, but the manuscript needs to be improved substantially prior to publication.
General comments:
- simple summary must be improved. It has to be striking and draw the reader attention.
- abstract has been interestingly improved. Please, every P in italic. Change conclusion into something new for this breed. Otherwise, the data you are providing has been largely described.
- there still some M&M issues unclear
- you do need to deep in the season characteristics (temperature, humidity, rain, etc.)
- figures must follow the same style. Now they are difficult to follow with that much *, letters and numbers (I guess they are mean values)
- results, in general, are difficult to follow
- they improved English style, but it does still needing improvings
- please, do not send the manuscript like this. Send it always clean with changes highlighted (different colors, one for each reviewers). Now it is difficult to read and I do not have clear what figures still and which ones are deleted. As well, you left also bubbles in Lithuanian… please take care of this
- I encourage authors to focus discussion on the breed. Because this kind of studies are widely performed. The new data comes from this breed, so please, rewrite and discuss depending on this breed.
Some other tips:
Ln 3. a Biomarker for Early Diagnosis of Hyperketonemia. Delete this from the title
Ln 59-61. Rewrite.
Ln 171-172. 15.73% higher, lower??? You use this kind of description in some other points, please clarify
Ln 220-221. Where this equation comes from?
Author Response

(The authors gave the same response as above.)

Round 3
Reviewer 1 Report
Thanks for your response.
Author Response
Hello,
thank you for your review.
Reviewer 2 Report
Authors included the suggestions that I proposed. I think that the manuscript has been improved substantially but I would encourage authors to keep working and to include the suggestions I propose:
Ln 51-53. Rewrite.
Ln 60. Add reference.
Ln 63. Ref is not placed properly
Ln 65. The same as ln 63
Ln 70-71. Add reference. Unique?
Ln 77. Add reference
Ln 83 1,200
Ln 98. 9,613
Ln 101. The same as ln 63
Ln 110. The same as ln 63
Ln 133. Delete space before .
Ln 137. Delete the
Ln 139. Please always the same number of decimals. I think 2 is enough. Apply this throughout the manuscript.
Ln 141. The same as ln 83
Ln 144. Delete the
Ln 150. 6.74% higher/lower in XX than XX
Ln 166-174. Too much information and confusing. I would add a table or reduce. Maybe compare 1st vs 2nd vs ≥3rd, it could help to enhance clarity
Ln 182. Avoid double spacing
Ln 183. P in italic. Check the whole manuscript
Ln 184-189. The same as ln 166-174
Table 1.
- 2 decimals
- include a p-value column to enhance clarity
Figure 2. I would have used the same format for every figure
Ln 234 and 250. Apply same comments than for table 1
Ln 320. Avoid contractions in formal English
References have no DOI, please include them all.
Author Response
Hello,
thank you for your review.
